# The Novel Pimavanserin Derivative ST-2300 with Histamine H_3_ Receptor Affinity Shows Reduced 5-HT_2A_ Binding, but Maintains Antidepressant- and Anxiolytic-like Properties in Mice

**DOI:** 10.3390/biom12050683

**Published:** 2022-05-10

**Authors:** Karthikkumar Venkatachalam, Sicheng Zhong, Mariam Dubiel, Grzegorz Satała, Bassem Sadek, Holger Stark

**Affiliations:** 1Department of Pharmacology & Therapeutics, College of Medicine and Health Sciences, United Arab Emirates University, Al Ain P.O. Box 17666, United Arab Emirates; karthikkumar@uaeu.ac.ae; 2Zayed Center for Health Sciences, United Arab Emirates University, Al Ain P.O. Box 17666, United Arab Emirates; 3Institute of Pharmaceutical and Medicinal Chemistry, Heinrich Heine University Düsseldorf, Universitaetsstr. 1, 40225 Dusseldorf, Germany; sicheng.zhong@uni-duesseldorf.de (S.Z.); mariam.dubiel@hhu.de (M.D.); 4Department of Medicinal Chemistry, Maj Institute of Pharmacology, Polish Acadamy of Sciences, 12 Smetna Street, 31-343 Krakow, Poland; satala@if-pan.krakow.pl

**Keywords:** histamine H_3_ receptor, 5-HT_2A_ receptor, pimavanserin, ACP-103, fluoxetine, antidepressant, anxiolytic, forced swim test, tail suspension test, open field test

## Abstract

The therapy of depression is challenging and still unsatisfactory despite the presence of many antidepressant drugs on the market. Consequently, there is a continuous need to search for new, safer, and more effective antidepressant therapeutics. Previous studies have suggested a potential association of brain histaminergic/serotoninergic signaling and antidepressant- and anxiolytic-like effects. Here, we evaluated the in vivo antidepressant- and anxiolytic-like effects of the newly developed multiple-active ligand ST-2300. ST-2300 was developed from 5-HT_2A/2C_ inverse agonist pimavanserin (PIM, ACP-103) and incorporates a histamine H_3_ receptor (H_3_R) antagonist pharmacophore. Despite its parent compound, ST-2300 showed only moderate serotonin 5-HT_2A_ antagonist/inverse agonist affinity (*K*_i_ value of 1302 nM), but excellent H_3_R affinity (*K*_i_ value of 14 nM). In vivo effects were examined using forced swim test (FST), tail suspension test (TST), and the open field test (OFT) in C57BL/6 mice. Unlike PIM, ST-2300 significantly increased the anxiolytic-like effects in OFT without altering general motor activity. In FST and TST, ST-2300 was able to reduce immobility time similar to fluoxetine (FLX), a recognized antidepressant drug. Importantly, pretreatment with the CNS-penetrant H_3_R agonist (*R*)-α-methylhistamine reversed the antidepressant-like effects of ST-2300 in FST and TST, but failed to reverse the ST-2300-provided anxiolytic effects in OFT. Present findings reveal critical structural features that are useful in a rational multiple-pharmacological approach to target H_3_R/5-HT_2A_/5-HT_2C_.

## 1. Introduction

Nearly four decades since its discovery, the histamine H_3_ receptor (H_3_R) has become a widely accepted drug target for multiple central nervous disorders. H_3_R is mainly localized in the central nervous system. The function as an autoreceptor is based on a blockage of Ca^2+^ current, which is part of histaminergic neurons’ pacemaker function, controlling firing, histamine synthesis, and release [1]. As heteroreceptors, H_3_R controls the release of several other neurotransmitters from GABAergic, glutamatergic, serotonergic, noradrenergic, cholinergic, and dopaminergic neurons in the mammalian brain [2,3].

To date, the only marketed selective H_3_R antagonist/inverse agonist is Pitolisant (Wakix^®^), which demonstrated a pronounced wake-promoting effect and is currently used in the EU and the U.S. for the therapeutic management of narcolepsy with or without cataplexy. Additionally, in a recent trial to treat obstructive sleep apnoea, pitolisant has shown excellent clinical outcomes in ameliorating the excessive daytime sleepiness leading to EMA approval (Ozawade^®^) [4].

Many H_3_R ligands have demonstrated attractive benefits in various preclinical models for neuropsychiatric and neurodegenerative diseases, including cognitive deficits [5,6], attention deficit hyperactivity disorder [7,8,9], autistic behavior [10,11,12], epilepsy [13,14], and depression [15,16,17]. Though the clinical significance is still unclear, high-level expression of H3R in the prefrontal cortex of schizophrenia patients suggested an involvement of the H_3_R in schizophrenia [18]. However, no breakthrough has been achieved by H_3_R ligands so far in the field of depression and schizophrenia.

Pimavanserin (PIM, ACP-103, Nuplazid^®^) is an atypical antipsychotic drug approved by the FDA in 2016 for the treatment of hallucinations and delusions associated with Parkinson’s disease psychosis. PIM acts as an inverse agonist at 5-HT_2A_ and 5-HT_2C_, where the affinity for 5-HT_2A_ is higher than that for 5-HT_2C_. No striking off-target binding to a series of other G-protein coupled receptors (GPCRs), including H_3_R, ion channels, or transporters was reported [19]. Moreover, in a Phase II clinical trial, PIM has shown potential efficacy in treating psychosis associated with Alzheimer’s disease [20]. In a further Phase II clinical trial (CLARITY study, ClinicalTrials.gov Identifier: NCT03018340), PIM has shown robust effects in major depression disorder (MDD) patients who did not properly respond to first- and second-generation antidepressants [21]. Additionally, an improved condition of insomnia and daytime sleepiness has been observed [22]. PIM has a generally good safety and tolerability profile. Only a low rate of typical side effects of antidepressants and antipsychotics, such as daytime sleepiness, weight gain, sexual dysfunction, extrapyramidal symptoms, and metabolic disorders have been observed.

PIM may have further advantages in treating MDD. Evidence suggests that the 5-HT_2A_ and 5-HT_2C_ receptors may be associated with suicidal behavior in depressive patients [23,24]. However, subjects with previous suicide attempts were excluded from the CLARITY study. The benefit regarding suicidal ideation could not be determined in this subpopulation. However, further studies are still required to gain a firm conclusion [25]. Therefore, simultaneous antagonism of H_3_R and 5-HT_2A/2C_ may provide synergistic effects in the treatment of depression and psychiatric disorders with additional benefits, such as suicidal ideation reduction, wake-promoting, and pro-cognitive properties.

Following the concept of developing multiple-acting ligands with presumably more efficient therapeutic effects than selective agents, the multiple-active ligand ST-2300 was designed, comprising a modified structure of PIM and the H_3_R antagonist pharmacophore *N*-(3-phenoxypropyl)piperidine (Figure 1) (for review see references [26,27,28,29,30,31]). In this work, we present the synthesis of ST-2300, its in vitro characterization at various histaminergic, dopaminergic, and serotonergic receptors, as well as the in vivo evaluation, in preclinical animal models, of depression- and anxiety-like behaviors, namely the forced swim test (FST), tail suspension test (TST), and open field test (OFT).

## 2. Materials and Methods

### 2.1. Chemistry

Compound ST-2300 was synthesized according to the procedure described for PIM in a patent of ACADIA [32], from isocyanate **4** and secondary amine **5**. Instead of the gaseous phosgene applied in the patent, liquid diphosgene was used to prepare the isocyanate derivative **4** from the benzylamine derivative **3**, which was obtained from a series of nucleophilic substitutions and hydrogenation reactions (Figure 2). 

#### 2.1.1. General Notes

Nuclear magnetic resonance (NMR) spectroscopy: Avance III 75, 300 MHz (Bruker, Reinstädten, Germany). Data for proton NMR are reported in the following order: multiplicity (br, broad; d, doublet; m, multiplet; p, quintet; s, singlet; and t, triplet; q, quartet; p, quintet), approximate coupling constants *J* expressed in Hertz (Hz), and the number of protons. Mass spectrometry: atmospheric pressure chemical ionization (APCI+/−) on Advion Expression L CMS (Advion, Ithaca, NY, USA). Liquid chromatography coupled with mass spectrometry (LC-MS): LC system: Elute SP (HPG 700) (Bruker Daltronics) equipped with a vacuum degasser, autosampler, column heater; column: Intensity Solo 2 C18 RP column (100 mm × 2.1 mm); MS-system: amaZon speed ETD ion trap LC/MSn system (Bruker Daltronics), equipped with electrospray; nebulizer: nitrogen, 15 Psi; dry gas: nitrogen, 8 L/min, 200 °C; mass range mode: Ultrascan. Alternating ion-polarity has been turned on, scan range: *m*/*z*: 80–1200. Sample preparation: the stock solutions in MeOH (approximately 1 mg/mL) have been diluted with MeOH (LC-MS grade), and the concentrations of approximately 0.1–0.2 mg/mL were obtained. The volume of injection was 2 µL. The column temperature has been conditioned at 50 °C. The mobile phase consisted of gradients from acetonitrile (LC-MS grade, Sigma Aldrich, St. Louis, MO, USA) and H_2_O (LC-MS grade, Sigma Aldrich, St. Louis, MO, USA) with 0.1% of formic acid (FA) (*v*/*v*) (Merck KGaA, Darmstadt, Germany). The flow rate was 0.2 mL/min, the measurements were only analyzed in positive mode, and the relative purity of the compounds was determined. Eluent mixture: acetonitrile:H_2_O (*v*/*v*): 0–4 min. 95:5, 4–5 min. gradient to 90:10, 5–12 min gradient to 80:20, 12–13 min. 80:20, 13–15 min. gradient to 5:95, reconditioning: 15–17 min. gradient to 95:5. 

#### 2.1.2. Syntheses and Analyses

4-[3-(Piperidin-1-yl)propoxy]benzonitrile (**2**) [33]. 

The preparation of compound **2** was performed according to the following process: 4-Cyanophenol (2.62 g, 22 mmol), alkyl chloride **1**, whose synthesis was described previously [34] (3.96 g, 20 mmol), K_2_CO_3_ (16 g, 119 mmol), and KI (0.70 g, 4.2 mmol) were suspended in acetone 100 mL. The reaction mixture was heated at reflux for 18 h. Upon the completion of the reaction monitored via thin-layer chromatography (TLC), the inorganic residue was removed by filtration. The solution was concentrated and diluted with ethyl acetate 100 mL and washed with 1 M NaOH 60 mL. The wash process was repeated until all phenolic derivative was removed from the organic layer (monitored via TLC). The organic layer was washed with one portion of 50 mL H_2_O and one portion of 50 mL brine was subsequently dried with anhydrous MgSO_4_, and evaporated in a rotary evaporator. The product was obtained in a yield of 93% as a transparent brownish viscous liquid and used without further purification in the next steps. 

^1^H NMR (300 MHz, DMSO-*d^6^*) δ 7.74 (d, *J* = 8.9 Hz, 2H, Ph-2,6-*H*), 7.09 (d, *J* = 8.9 Hz, 2H, Ph-3,5-*H*), 4.07 (t, *J* = 6.4 Hz, 2H, -O-C*H_2_*-), 2.41–2.24 (m, 6H, overlap: Piperidine-C*H_2_*, Piperi-2,6-*H_2_*), 1.97–1.78 (m, 2H, -O-CH_2_-C*H_2_*-), 1.53–1.42 (m, 4H, Piperidine-3,5-*H_2_*), 1.41–1.31 (m, 2H, Piperidine-4-*H_2_*). Molecular weight calculated for chemical formula: C_15_H_20_N_2_O: 244.3 g/mol. APCI-MS (+): *m*/*z*: 245.2 [M + H^+^]^+^.

{4-[3-(Piperidin-1-yl)propoxy]phenyl}methanamine (**3**) [35].

Preparation of Raney-Nickel: 1.0 g of nickel–aluminium alloy was treated with 80 mL of 10% NaOH solution (aq. m/V), the mixture was heated at 90 °C for 60 min, and the colorless solution was decanted. The black-colored nickel particles were washed with H_2_O. The wash process was repeated until the pH value of the wash water was neutral. The nickel particles were subsequently washed three times with 20 mL of MeOH to remove the rest of the water and transferred cautiously to an autoclave tube. 

Hydrogenation with Raney-Nickel: the cyano-derivative **2** (4.5 g, 18.5 mmol) was dissolved in 100 mL MeOH (NH_3_), and this solution was given to the activated nickel particles. The reaction was stirred under 5 bar of hydrogen and heated at 40 °C. After 18 h, the pressure of the gas was released cautiously, the suspension of the reaction was filtered through kieselgur. The filter cake was rinsed three times with 20 mL of MeOH. The filtrate was collected and evaporated to dryness to obtain compound **3** in a yield of 81% as a yellowish transparent viscous liquid. The product was sufficiently pure to use in the next steps without further purification. 

^1^H NMR (300 MHz, CDCl_3_) δ 7.20 (d, *J* = 8.1 Hz, 2H, Ph-2,6-*H*), 6.86 (d, *J* = 8.0 Hz, 2H, Ph-3,5-*H*), 3.99 (t, *J* = 6.4 Hz, 2H, -O-C*H_2_*-), 3.82 (s, 2H, Ph-C*H_2_*-), 2.55–2.32 (m, 6H, overlap: Piperi-C*H_2_*-, Piperi-2,6-*H_2_*), 1.97 (p, *J* = 6.6 Hz, 2H, -O-CH_2_-C*H_2_*-), 1.78 (br, 2H, -N*H_2_*), 1.60 (p, *J* = 5.6 Hz, 4H, Piperi-3,5-*H_2_*), 1.50–1.37 (m, 2H, Piperi-4-*H_2_*). Molecular weight calculated for chemical formula: C_15_H_24_N_2_O: 248.4 g/mol. APCI-MS (+): *m*/*z*: 249.2 [M + H^+^]^+^.

1-{3-[4-(Isocyanatomethyl)phenoxy]propyl}piperidine hydrochloride (**4**).

The isocyanate derivative **4** was synthesized via the following method: a solution of the benzylamine derivative **3** (0.28 g, 1.12 mmol) and triethylamine (0.56 mL, 4 mmol) in 5 mL of dichloromethane was given to a solution of diphosgene (0.12 mL, 1 mmol) in 5 mL of dichloromethane at 0 °C. The mixture was allowed to reach r.t. and further stirred for 1.5 h. The solvent was subsequently removed in the rotary evaporator. The crude product was yielded quantitatively as a white solid and used in the next step without further purification. 

^1^H NMR (300 MHz, CDCl_3_) δ 7.19 (d, *J* = 8.6 Hz, 2H, Ph-2,6-*H*), 6.83 (d, *J* = 8.7 Hz, 2H, Ph-3,5-*H*), 4.33 (d, *J* = 5.9 Hz, 2H, Ph-C*H_2_*-), 4.01 (t, *J* = 5.9 Hz, 2H, -O-C*H_2_*-), 2.89–2.71 (m, 6H, overlap, Piperi-C*H_2_*-, Piperi-2,6-*H_2_*), 2.29–2.14 (m, 2H, -O-CH_2_-C*H_2_*-), 1.86 (p, *J* = 5.7 Hz, 4H, Piperi-3,5-*H_2_*), 1.63–1.51 (m, 2H, Piperi-4-*H_2_*). Molecular weight calculated for chemical formula: C_16_H_2_N_2_O_2_: 274.4 g/mol. APCI-MS (+): *m*/*z*: 275.2 [M + H^+^]^+^.

*N*-(4-Fluorobenzyl)-1-methylpiperidin-4-amine (**5**) [32].

Compound **5** was synthesized via Borch reductive amination: 4-fluorobenzylamine (0.50 g, 4 mmol), 1-methylpiperidin-4-one (0.45 g, 4 mmol), and glacial acetic acid (0.34 mL, 6 mmol) were dissolved in 20 mL of dichloroethane. The reaction mixture was stirred in r.t. for 1 h. NaBH(CH_3_COO)_3_ (1.00 g, 4.7 mmol) was added to the aforementioned solution, and the reaction was further stirred at r.t. for 18 h. Upon the completion of the reaction, monitored via TLC, 50 mL of NaHCO_3_ solution was given to the mixture. The resulting mixture was extracted with three portions of 50 mL of DCM. The combined organic layer was dried with anhydrous MgSO_4_ and concentrated in a rotary evaporator. The residue was purified with column chromatography, where the solvent mixtures of DCM and MeOH (NH_3_) in ratios of 98:2 and 95:5, respectively, were used as eluents. The product was obtained in a yield of 43% as a yellow-colored liquid. 

^1^H NMR (300 MHz, CDCl_3_) δ 7.34–7.22 (m, 2H, Ph-2,6-*H*), 7.07–6.93 (m, 2H, Ph-3,5-*H*), 3.78 (s, 2H, Ph-C*H_2_*-), 2.87–2.74 (m, 2H, Piperidine-2,6-*H_eq_*), 2.58–2.40 (m, 1H, Piperidine-4-*H*), 2.26 (s, 3H, -C*H_3_*), 2.06–1.82 (m, 4H, Piperidine-2,6-*H_ax_*, Piperi-3,5-*H_eq_*), 1.53–1.34 (m, 2H, Piperidine-3,5-*H_ax_*). Molecular weight calculated for chemical formula: C_13_H_19_FN_2_: 222.3 g/mol. APCI-MS (+): *m*/*z*: 223.0 [M + H^+^]^+^.

1-(4-Fluorobenzyl)-1-(1-methylpiperidin-4-yl)-3-{4-[3-(piperidin-1-yl)propoxy]benzyl}urea (**ST-2300**).

The PIM analog **ST-2300** was synthesized in analogy to the methods patented previously by ACADIA [32]. A solution of the isocyanate derivative **4** (0.31 g, 1.12 mmol) in 10 mL of tetrahydrofuran was given to a solution of compound **5** (0.32 g, 1.44 mmol) in 10 mL tetrahydrofuran. The reaction mixture was stirred at r.t. for 18 h. The solvent was subsequently reduced in the rotary evaporator. The residue was purified with column chromatography, where the solvent mixture of DCM and MeOH in a ratio of 90:10 and the mixture of DCM and MeOH (NH_3_) in a ratio of 90:10 were used as eluents. ST-2300 was obtained in a yield of 66% as a white to beige solid with a melting point of 114.0 °C. 

^1^H NMR (300 MHz, CDCl_3_) δ 7.21–7.10 (m, 2H, 4-F-Ph-2,6-*H*), 7.05–6.89 (m, 4H, 4-O-Ph-2,6-*H*, 4-F-Ph-3,5-*H*), 6.76 (d, *J* = 8.7 Hz, 2H, 4-O-Ph-3,5-*H*), 4.47 (t, *J* = 5.4 Hz, 1H, -CO-N*H*-), 4.36–4.27 (m, 3H, overlap: 4-F-Ph-C*H_2_*-, 1-Me-Piperidine-4-C*H*-), 4.25 (d, *J* = 5.4 Hz, 2H, 4-O-Ph-C*H_2_*-), 3.94 (t, *J* = 6.4 Hz, 2H, -O-C*H_2_*-), 2.90–2.77 (m, 2H, 1-Me-Piperidine-2,6-*H_eq_*), 2.49–2.33 (m, 6H, overlap: Piperidine-C*H_2_*-, Piperi-2,6-*H_2_*), 2.23 (s, 3H, -C*H_3_*), 2.04 (td, ^3,4^*J* = 11.6, 3.2 Hz, 2H, 1-Me-Piperidine-2,6-*H_ax_*), 1.99–1.88 (m, 2H, -O-CH_2_-C*H_2_*-), 1.77–1.62 (m, 4H, 1-Me-Piperidine-3,5-*H_2_*), 1.57 (p, J = 5.2 Hz, 4H, Piperidine-3,5-*H_2_*), 1.47–1.36 (m, 2H, Piperidine-4-*H_2_*). ^13^C NMR (75 MHz, CDCl_3_) δ 162.06, 158.29, 158.15, 134.20, 131.38, 128.71, 127.75, 115.79, 114.59, 66.60, 56.06, 55.35, 54.71, 52.31, 46.17, 45.18, 44.49, 30.26, 26.88, 26.03, 24.49. Molecular weight calculated for chemical formula: C_29_H_41_FN_4_O_2_: 496.7 g/mol. Purity determined via LC-MS: 96.18%, *m*/*z*: 249.08 [(M + 2H^+^)/2]^+^ and 497.33 [M + H^+^]^+^.

### 2.2. In Vitro Characterization

#### 2.2.1. In Vitro Radioligand Displacement Assays at Different GPCRs

##### Histamine H_3_ Receptor

Radioligand displacement assays were performed as described previously [31,36]. HEK293 cells stably expressing the human histamine H_3_ receptor [37] were grown in Dulbecco’s modified Eagle’s medium, supplemented with 2 mM glutamine, 10 mM HEPES, 10% fetal bovine serum, and 10 μL/mL penicillin/streptomycin in an atmosphere of 5% CO_2_ at 37 °C. Cells were grown to confluence. The medium was removed, and cells were collected in 15 mL ice-cold phosphate-buffered saline (PBS) (140 mM NaCl, 3 mM KCl, 1.5 mM KH_2_PO_4_, 8 mM Na_2_HPO_4_, pH 7.4). The cell suspension was centrifuged two times at 3000× *g* for 10 min with subsequent aspiration in 20 mL of PBS buffer at 4 °C. After the last centrifugation step, cells were disrupted and homogenized using an ULTRA-TURRAX^®^ T25 digital (IKA, Staufen, Germany) in ice-cold binding buffer (12.5 mM MgCl_2_, 100 mM NaCl, and 75 mM Tris/HCl, pH 7.4). The cell membrane homogenate was centrifuged two times at 20,000× *g* for 20 min at 4 °C and homogenized again by sonication at 4 °C, and kept in ice-cold binding buffer. The membranes were stored at −80 °C until use. 

Competition binding experiments were carried out as follows: membranes (20 μg/well in a final volume of 0.2 mL binding buffer) were incubated with [^3^H]-*N^α^*-methylhistamine (2 nM), purchased from PerkinElmer (Waltham, MA, USA), and different concentrations of the respective test ligand. Assays were run at least in duplicates with eleven appropriate concentrations between 0.01 nM and 100 μM of the test compound. The preparation of competitor concentrations was carried out by serial dilution of 10 and 3 mM stock solution using a Freedom EVO pipetting instrument (TECAN^®^, Männerdorf, Switzerland). Incubations were performed for 90 min at r.t. under shaking. Non-specific binding was determined in the presence of the selectively acting histamine H_3_ receptor inverse agonist/antagonist pitolisant (10 μM). The bound radioligand was separated from free radioligand by filtration through GF/B filters pretreated with 0.3% (*m*/*v*) polyethylenimine using an Inotech cell harvester (Dottikon, Switzerland). Unbound radioligand was removed by three washing steps with approximately 0.25 mL/well of cold deionized water. Liquid scintillation counting using a PerkinElmer MicroBeta Trilux scintillation counter was used to determine the amount of radioactivity collected on the filter used in the current experiment. The competition binding data were analyzed by the software GraphPad Prism™ (2016, version 7.01, San Diego, CA, USA) using a non-linear regression fit “one-site competition”. The inhibitory constant (*K*_i_) was calculated from the *IC*_50_ values according to the Cheng-Prusoff equation [38], expressed as the means from at least three independent experiments in duplicate with 95% confidence intervals. 

##### Histamine H_1_ Receptor

To determine the human H_1_R affinity, compounds and reference substances were tested in a [^3^H]-pyrilamine displacement assay. The assay was conducted with membrane preparations of CHO-K1 cells stably expressing hH_1_R. Transfection, cell culture, and membrane preparation were described in detail by Smit and colleagues [39]. For the competition binding assay, membrane fractions (40 μg/well in a final volume of 0.2 mL binding buffer) were incubated with [^3^H]-pyrilamine (1 nM) and the test compounds in different concentrations for 120 min at r.t. under shaking. The experimental assays were carried out in triplicate with at least six appropriate concentrations between 0.1 nM and 100 μM for the test compounds. Nonspecific binding was evaluated in the presence of the standard H_1_R antagonist chlorpheniramine hydrogen maleate at a concentration of 10 µM. The following steps were performed as described before. 

##### Histamine H_4_ Receptor

To determine the human H_4_R affinity, compounds and reference structures were examined in a [^3^H]-histamine dihydrochloride competition binding assay. Cell culture and membrane preparation of Sf9 cells co-expressing hH_4_R-Gα_i2_ fusion protein with Gβ_1_γ_2_ was prepared as described by Schneider and co-workers [40]. 

The competition binding assay has been performed as described previously by Sander, Kottke, and Schwed from the Stark Lab [41,42]. Cell membranes were sedimented by 3 min centrifugation at 25 °C and 130 rpm and were resuspended in binding buffer. Competition binding experiments were carried out by incubating the membranes (40 µg/0.2 mL binding buffer) and [^3^H]-histamine dihydrochloride (10 nM) with test compounds for 60 min at r.t. Assays were run in triplicates with four to seven appropriate concentrations between 0.1 nM and 100 µM for the test compound. Nonspecific binding was determined in the presence of 100 µM JNJ7777120. The following steps were performed as described before. 

##### Dopamine D_3_ and D_1_ Receptor

To determine the human D_3_R affinity, compounds and reference substances were tested in a [^3^H]-spiperone competition binding assay as described by Bautista-Aguilera et al. [43]. The assay was conducted with membrane preparations of CHO-K1 cells stably expressing hD_3_R. Membrane fractions (20 μg/well in a final volume of 0.2 mL binding buffer) were incubated with [^3^H]-spiperone (0.2 nM) and the test compounds for 120 min at r.t. under shaking. Assays were run in triplicates with at least six appropriate concentrations between 0.1 nM and 100 µM for the test compound. Nonspecific binding was determined in the presence of 10 µM haloperidol.

hD_1_R affinity was determined according to the procedure of hD_3_R [43]. Membrane preparations of HEK293 cells stably expressing the hD_1_R were used with 20 µg/well. [^3^H]-SCH23390 was used as radioligand in a concentration of 0.3 nM. Nonspecific binding was determined in the presence of 100 µM fluphenazine. The following steps were performed as describedbefore.

##### Dopamine D_2_ and Serotonin (5-HT_1A_, 5-HT_2A_, 5-HT_6_, and 5-HT_7_) Receptor

HEK293 cells with a stable expression of human 5-HT_1A_, 5-HT_6_, 5-HT_7b_, and D_2L_ (prepared with the use of Lipofectamine 2000), or CHO-K1 cells with plasmids containing the sequence coding for the human serotonin 5-HT_2A_ receptor (PerkinElmer), were maintained at 37 °C in a humidified atmosphere with 5% CO_2_ and grown in Dulbecco’s Modified Eagle Medium containing 10% dialyzed fetal bovine serum and 500 µg/mL G418 sulfate. For membrane preparation, cells were subcultured in 150 cm^2^ flasks, grown to 90% confluence, washed twice with prewarmed to 37 °C phosphate buffered saline (PBS) and pelleted by centrifugation (200× *g*) in PBS containing 0.1 mM EDTA and 1 mM dithiothreitol. Prior to membrane preparation, the pellets were stored at −80 °C. 

Cell pellets were thawed and homogenized in 10 volumes of assay buffer using an Ultra Turrax tissue homogenizer, and were then centrifuged twice at 35,000× *g* for 15 min at 4 °C, with incubations for 15 min at 37 °C in between. The composition of the assay buffers was as follows: for 5-HT_1A_: 50 mM Tris HCl, 0.1 mM EDTA, 4 mM MgCl_2_, 10 µM pargyline, and 0.1% ascorbate; for 5-HT_2A_: 50 mM Tris HCl, 0.1 mM EDTA, 4 mM MgCl_2_, and 0.1% ascorbate; for 5-HT_6_: 50 mM Tris HCl, 0.5 mM EDTA, and 4 mM MgCl_2_; for 5-HT_7b_: 50 mM Tris HCl, 4 mM MgCl_2_, 10 µM pargyline, and 0.1% ascorbate; for D_2_R: 50 mM Tris HCl, 1 mM EDTA, 4 mM MgCl_2_, 120 mM NaCl, 5 mM KCl, 1.5 mM CaCl_2_, and 0.1% ascorbate. All assays were incubated in a total volume of 200 µL in 96-well microtiter plates for 1 h at 37 °C, except for 5-HT_1A_ and 5-HT_2A_, which were incubated at room temperature and 27 °C, respectively. The process of equilibration was terminated by rapid filtration through Unifilter plates with a 96-well cell harvester and the radioactivity retained on the filters was quantified on a Microbeta plate reader (PerkinElmer, Waltham, MA, USA). For displacement studies, the assay samples contained as radioligands (PerkinElmer, Waltham, MA, USA) were: 2.5 nM [^3^H]-8-OH-DPAT (135.2 Ci/mmol) for 5-HT_1A_; 1 nM [^3^H]-ketanserin (53.4 Ci/mmol) for 5-HT_2A_; 2 nM [^3^H]-LSD (83.6 Ci/mmol) for 5-HT_6_; 0.8 nM [^3^H]-5-CT (39.2 Ci/mmol) for 5-HT_7_ or or 2.5 nM [^3^H]-raclopride (76.0 Ci/mmol) for D_2L_R. Non-specific binding was defined with 10 µM of 5-HT in 5-HT_1A_ and 5-HT_7_ binding experiments, whereas 20 µM of mianserin, 10 µM of methiothepine, or 10 µM of haloperidol were used in 5-HT_2A_, 5-HT_6_, and D_2L_ assays, respectively. Each compound was tested in triplicate at 7 concentrations (10^−10^–10^−4^ M). The inhibition constants (*K*_i_) were calculated from the Cheng-Prusoff equation [38]. Results are expressed as means of two separate experiments. 

### 2.3. In Vivo Assessment

#### 2.3.1. Animals

Inbred adult male C57BL/6 mice (aged 10–12 weeks and weighing 20–25 g) were obtained from the animal facility in the College of Medicine and Health Sciences, United Arab Emirates University. The animals were maintained in a separate air-conditioned room with controlled temperature and humidity (24 ± 2 °C and 55 ± 15%, respectively), with a 12 h light/dark cycle, and ad libitum to food and water. All the experiments were performed between 9.00 am and 3.00 pm. The procedures were approved by the Institutional Animal Ethics Committee of College of Medicine and Health Sciences/United Arab Emirates University (Approval No. ERA-2019-6010). To reduce the suffering of the animals, a minimum number of animals were used in this study, whereas the objectives were not compromised. 

#### 2.3.2. Drugs

The multiple-active test compound 1-(4-fluorobenzyl)-1-(1-methylpiperidin-4-yl)-3-{4-[3-(piperidin-1-yl)propoxy]benzyl}urea (ST-2300, 5, 10 and 15 mg/kg, i.p.) and the reference drug PIM (5, 10, and 15 mg/kg, i.p.) were provided by the Institute of Pharmaceutical and Medicinal Chemistry, Heinrich Heine University Düsseldorf, Germany. Diazepam (DZP, 1 mg/kg, i.p.), manufactured by Gulf Pharmaceutical Industries (Ras Al Khaimah, United Arab Emirates), was obtained from Dr. Essam Emam (Department of Medicine, Tawam Hospital, Al Ain, United Arab Emirates) and fluoxetine (Prozac^®^, 10 mg/kg) was obtained from Eli Lilly (Indianapolis, IN, USA). The brain-penetrant H_3_R agonist (*R*)-α-methylhistamine (RAM, 10 mg/kg, i.p) was purchased from Sigma-Aldrich (St. Louis, MO, USA). 

#### 2.3.3. Behavioral Studies

These experiments provide the first behavioral assessment of the ST-2300 in anxiety-like and depression-like behaviors in mice. All doses are expressed in terms of the free base of all test compounds. The drugs used in the current study were dissolved in saline and injected i.p. at a volume of 1 mL/kg. All experimental procedures were conducted in a blinded fashion in which the experimenter was uninformed about the specific treatment groups to which an animal group belonged. Saline, ST-2300 (5, 10, and 15 mg/kg), PIM (5, 10, and 15 mg/kg), FLX (10 mg/kg), RAM (10 mg/kg), and DZP (1 mg/kg) were injected i.p. 30–45 min before the beginning of the behavioral experiments. 

##### Forced Swim Test

The FST test was performed according to previously described protocols [16,44,45]. Mice were individually placed into glass cylinders containing 15 cm of water at ∼25 °C. The mice were left in the cylinders for 6 min and immobility was observed and analyzed. The mice were then removed from the container and left to dry in a heated enclosure before being returned to their home cages. The mice were considered to be immobile when they ceased struggling and remained floating motionless in the water (without any vertical or horizontal movements), making only the movements necessary to keep their heads above the water level as described previously [16,45]. 

##### Tail Suspension Test

This experimental protocol was followed as previously described, however with slight modifications [16]. Each mouse was suspended on the edge of a rod 50 cm above a table top using adhesive Scotch tape placed approximately 1 cm from the tip of the tail. Tail climbing was prevented by passing the mouse’s tail through a small plastic cylinder prior to suspension, as described by Can et al. [46]. The duration of immobility was manually scored for a 6 min observation period. Mice were considered immobile only when they hung down passively and were completely motionless. The parameter recorded was the number of seconds spent immobile. 

##### Open Field Test

To analyze the effect of ST-2300 on locomotion and anxiety behaviors in animals, OFT was carried out. In addition to locomotor activity, the OFT is usually used to measure anxiety-like behaviors in experimental rodents [10,16,47,48]. The test provides a unique opportunity to systematically assess novel environment exploration, general locomotor activity, and provide an initial screen for anxiety-related behavior in experimental rodents. The OF box consisted of a square box (45 × 45 × 30 cm). A 23 × 23 cm area in the center region was defined as the central arena, and the remaining area was defined as periphery area. The mice with a higher degree of anxiety prefer to stay closer to the walls of the box and spend less time in the center, whereas increased time spent in the central area indicates low anxiety level and high exploratory behaviors. The first 5 min of the experiment were considered as habituation, followed by 10 min test session. Total distance moved in the whole arena, time spent in the center, and in the periphery were recorded for 10 min using a charge-coupled, camera-assisted, motion-tracking apparatus and software (EthoVision 3.1, Noldus Information Technology, Wageningen, The Netherlands). After each mouse completed the test, the OF chamber was cleaned thoroughly with 70% (volume/volume; vol/vol) alcohol.

#### 2.3.4. Statistics

For behavioral assessments, data are expressed as means ± SEM. The data were analyzed for normality by assessing the sample distribution or skewness (−1.5 to +1.5 was considered as normally distributed). After the results had passed the tests for normality, the effects of drug treatments were analyzed by a two-way analysis of variance (ANOVA) and post hoc comparisons were performed with the Tukey’s test in case of a significant main effect. For statistical comparisons, the software package SPSS 25.0 (IBM Middle East, Dubai, United Arab Emirates) was used. The *p* values of less than 0.05 were considered as statistically significant. 

## 3. Results

For the pharmacological in vitro evaluation of ST-2300, competitive radioligand binding assays at various GPCRs were performed (Table 1). Our study confirmed the high receptor binding affinity of PIM to the 5-HT_2A_ receptor with a *K*_i_ value in the low nanomolar range. It should be noted that a *K*_i_ value in the picomolar concentration range, reported from Vanover et al. [19], could not be reproduced under our assay conditions. Furthermore, PIM demonstrated a good binding affinity at the 5-HT_6_ receptor and weaker binding at the dopamine D_2_ receptor. It was previously reported [19] that PIM does not show appreciable activity at dopaminergic/histaminergic GPCRs and lacks D_2_R antagonistic activity [49], which may explain the reduced side effect profile of PIM. Under our assay conditions, a good 5-HT_6_ affinity and a weak D_2_R affinity were observed. 

ST-2300 has shown excellent binding affinity at H_3_R (Table 1). However, despite the high structural similarity to PIM, ST-2300 displayed only a moderate affinity at 5-HT_2A_. Screenings performed on other monoamine receptors subtypes revealed no notable off-target affinities for ST-2300. 

In vivo antidepressant- and anxiolytic-like effects of ST-2300 were evaluated using FST, TST, and OFT in adult male C57BL7/6 mice. FST and TST were used to determine the antidepressant-like effects, while OFT was used to determine anxiolytic-like effects. Mice were pretreated with ST-2300 (0–15 mg/kg, intraperitoneal, (i.p.)) or PIM (0–15 mg/kg, i.p.). Fluoxetine (FLX, 10 mg/kg, i.p.) and diazepam (DZP, 1 mg/kg, i.p.) were used as reference compounds in the depressive-like and anxiety-like animal model, respectively. To determine whether the observed effects were associated with antagonism of H_3_Rs, (*R*)-α-methyl histamine (RAM, 10 mg/kg, i.p.) was co-administrated with ST-2300. To exclude any confounding effects of the CNS-penetrant H_3_R agonist RAM, RAM was administered to the control mice. 

### 3.1. Forced Swim Test (FST)

The results observed in FST revealed a significant and dose-dependent decrease of immobility time for mice pretreated with ST-2300 or PIM (Figure 3). Indeed, the decrease of immobility time for mice treated with ST-2300 (10 or 15 mg/kg) was not significantly different from the decrease of immobility time for mice treated with the reference drug PIM (10 or 15 mg/kg). The co-administration of the CNS-penetrant H_3_R agonist RAM with ST-2300 led to a significant increase of immobility time (Figure 4), demonstrating the reversal of the ST-2300-provided effects on immobility time with the co-administration of the H_3_R agonist. The latter conclusion shed light on the involvement of histaminergic neurotransmission mediated through H_3_Rs. 

### 3.2. Tail Suspension Test (TST) 

The results obtained in the TST show the effects of ST-2300 on depression-like behavior. Acute systemic pretreatment with ST-2300 (10 or 15 mg/kg, i.p.) significantly reduced the immobility time of tested mice as compared to control mice injected with saline (Figure 5). Interestingly, there is a remarkable difference between treatment with ST-2300 (10 or 15 mg/kg) or PIM (5–15 mg/kg). Accordingly, pretreatment with PIM failed to affect the overall immobility time when compared to control mice. Similar to the results observed for the FST, the effect of the ST-2300-induced reduction of immobility time was reversed by systemic co-administration with the CNS-penetrant H_3_R agonist RAM (Figure 6). 

### 3.3. Open Field Test

To test the effect of ST-2300 on anxiety-like behavior, an open field test (OFT) was performed. Pretreatment with ST-2300 or PIM did not alter the number of total line crossings (Figure 7A), excluding any effects of ST-2300 on locomotor activity. However, systemic ST-2300 pretreatment led to a dose-dependent increase of time spent in the center of the arena (Figure 7B), indicating the low anxiety level of mice when pretreated with ST-2300 (10 or 15 mg/kg, i.p.). On the contrary, the reference drug PIM (5–15 mg/kg, i.p.) failed to provide any appreciable effects on the anxiety levels in the assessed mice. Notably, the significant increase of time spent in the center following pretreatment with ST-2300 could not be reversed by co-administration with RAM, suggesting that histaminergic neurotransmission by H_3_Rs is not involved in the ST-2300-provided anxiolytic effects (Figure 7C). 

## 4. Discussion

With the novel designed multiple-active ligand ST-2300, we present a selective and simultaneously targeting H_3_R/5-HT_2A_ ligand with an innovative option for treating depressive and psychotic disorders. The linkage of PIM and the histamine H_3_R pharmacophore is a promising attempt to combine the antipsychotic properties of 5-HT_2A/2C_ antagonists/inverse agonists with the pro-cognitive effects of H_3_R antagonists/inverse agonists. The presented synthesis route is easily adaptable for further lead optimization. As a prototype of its class, ST-2300 already showed high receptor binding affinity at H_3_R and decent activity at 5-HT_2A_. It is known for PIM, which displays selective affinity at 5-HT_2A_ over 5-HT_2C_ [19]. For further investigation of ST-2300, the 5-HT_2C_ affinity should be determined. ST-2300 was lacking any appreciable off-target affinity, which was also previously reported for the reference drug PIM [19]. Though, with our assay conditions, PIM displayed a good affinity at 5-HT_6_ and a weak affinity at D_2_R. Since 5-HT_6_ antagonists are already under investigation for cognitive improvements in patients diagnosed with Alzheimer’s disease (AD) [50], this might contribute to a potential use of PIM and its derivatives for AD. 

Interestingly, the observed in vivo results showed that ST-2300 was superior or comparable to the reference drug PIM, using depressive-like and anxiety-like animal models. Accordingly, ST-2300 provided antidepressant-like effects in the FST as well as in the TST. In contrast to PIM, ST-2300 was capable of increasing the time spent in the center of the arena in the OFT, without altering the overall locomotor activity of the tested mice (overall line-crossing events were not influenced by the doses used). Importantly, the CNS-penetrant H_3_R agonist RAM reversed the antidepressant-like effects provided by ST-2300 in both paradigms assessed in FST and TST, indicating that the antagonism of H_3_Rs was responsible for the provided effects of ST-2300 in tested animals. However, RAM failed to reverse the anxiolytic-like effects provided by ST-2300 in OFT, indicating that neurotransmitters other than histamine are involved in the ST-2300-provided effects on anxiety levels of assessed mice in the OFT. Notably, previous reports in experimental rodents showed antidepressant- or anxiolytic-like effects of several H_3_R antagonists/inverse agonists (e.g., ST-1283, p*K*_i_ 9.62) [15,16,17]. However, the exact mechanisms of H_3_R antagonists regarding their anxiolytic and antidepressant effects remain to be fully elucidated. Further experimental efforts are necessary to fully understand the mechanisms of histaminergic and/or other monoaminergic action involved in anxiety and depression. 

In order to comprehend the therapeutic potential of the concept leading to ST-2300 applications, further approaches should focus on multiple-active ligands with a balanced H_3_R/5-HT_2A_ affinity. Moreover, the potential antipsychotic activity of ST-2300 still needs to be assessed in a battery of animal models to be able to generalize the current conclusions. 

## 5. Conclusions

Taken together, the novel multiple-active ligand ST-2300 with H_3_R and 5-HT_2A_ antagonist/inverse agonist affinities showed promising H_3_R-mediated in vivo antidepressant- as well as anxiolytic-like effects in tested animals. Therefore, multiple-active ligands with H_3_R affinity might address the need for safe and effective drugs with potential to be used in the treatment of depression- and anxiety-related neuropsychiatric disorders.

## Figures and Tables

**Figure 1 biomolecules-12-00683-f001:**
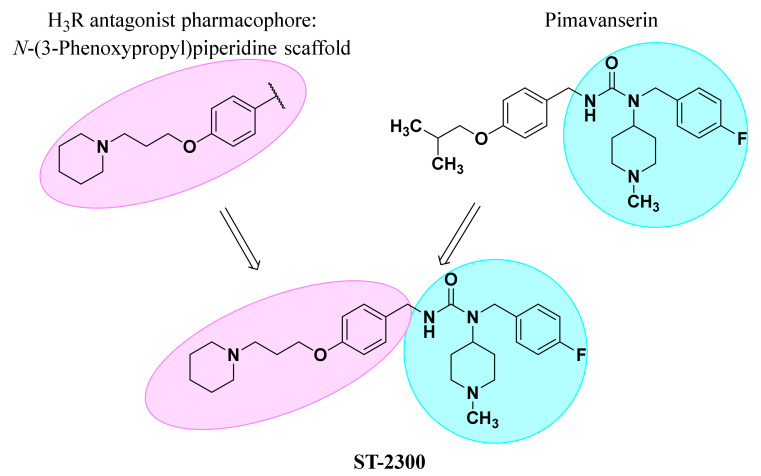
Design of multiple-active compound ST-2300.

**Figure 2 biomolecules-12-00683-f002:**
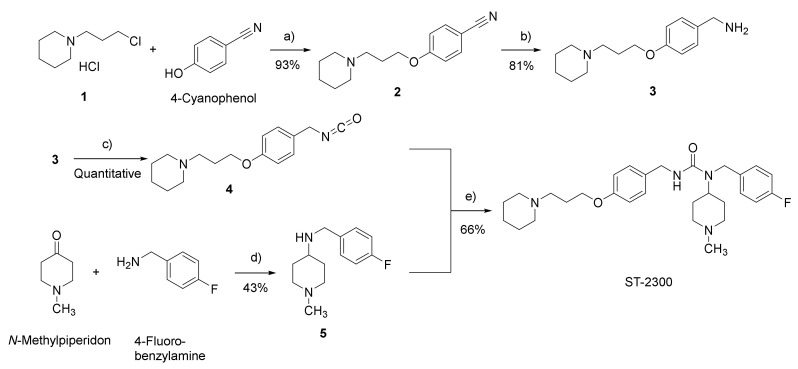
The synthesis of compound ST-2300. (**a**): K_2_CO_3_, KI (catalytic amount), acetone, reflux, 18 h; (**b**): Raney-Nickel, NH_3_ (saturated in MeOH), H_2_, 5 bar, 40 °C, 18 h; (**c**): Diphosgene, triethylamine, dichloromethane, 0 °C—room temperature, 2 h; (**d**): CH_3_COOH, NaBH(CH_3_COO)_3_, 1,2-dichloroethane, room temperature 18 h; (**e**): Tetrahydrofuran, room temperature, 18 h.

**Figure 3 biomolecules-12-00683-f003:**
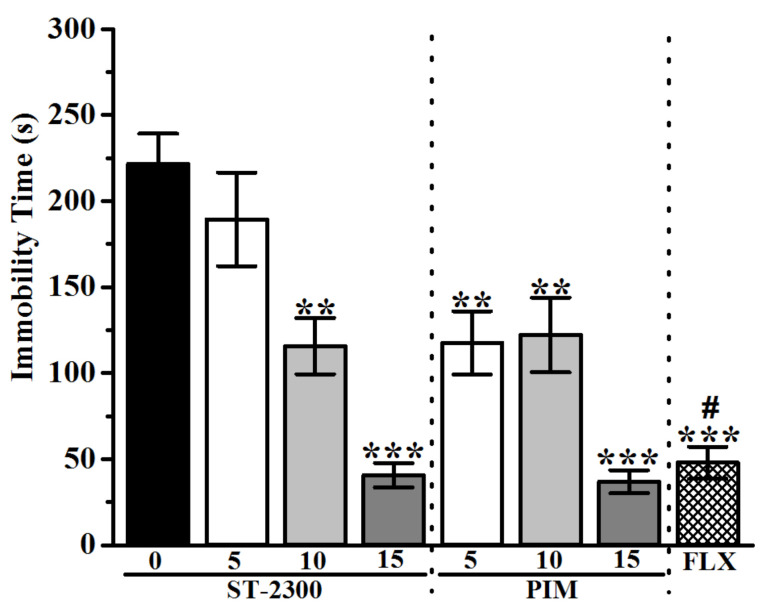
Effects of acute ST-2300 pre-treatment on depression-like behavior in FST. Acute ST-2300 dose-dependently decreased the immobility time in the FST. ** Denotes significant differences between drug doses and saline controls (*p* < 0.01); *** Denotes significant differences between drug doses and controls (*p* < 0.001). # Denotes significant differences between FLX and ST-2300 (5 and 10 mg/kg, i.p.) as well as PIM (5 and 10 mg/kg, i.p.) (*p* < 0.05). Vehicle (n = 9); ST-2300 (5 mg/kg, i.p.) (n = 7); ST-2300 (10 mg/kg, i.p.) (n = 6); ST-2300 (15 mg/kg, i.p.) (n = 7); PIM (5 mg/kg, i.p.) (n = 6); PIM (10 mg/kg, i.p.) (n = 7); PIM (15 mg/kg, i.p.) (n = 6); FLX (10 mg/kg, i.p.) (n = 9). Data are shown as the mean ± standard error of the mean. Abbreviations: PIM, pimavanserin; FLX, fluoxetine; FST, forced swim test.

**Figure 4 biomolecules-12-00683-f004:**
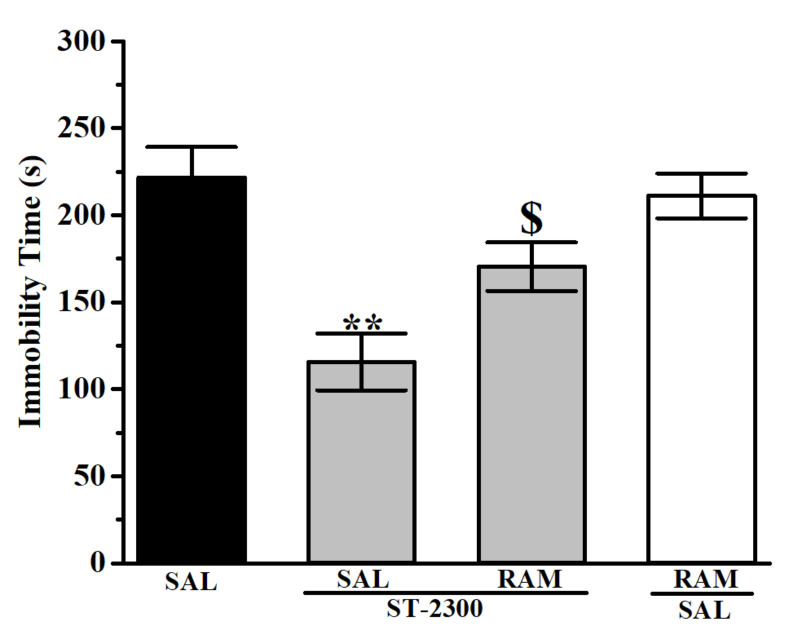
The H_3_R agonist RAM reversed the ST-2300-provided antidepressant-like effects in FST. Acute ST-2300 (10 mg/kg, i.p.) significantly decreased the immobility time in the FST. ** Denotes significant differences between ST-2300 (10 mg/kg, i.p.) and saline-treated control mice group (*p* < 0.01); $ Denotes significant difference when compared to ST-2300 (10 mg/kg, i.p.)-treated mice (*p* < 0.05). Vehicle (n = 9); ST-2300 (10 mg/kg) (n = 6); ST-2300 (10 mg/kg) co-administered with RAM (10 mg/kg) (n = 6); RAM (10 mg/kg) co-administered with vehicle (n = 9). Data are shown as the mean ± standard error of the mean. Abbreviations: RAM, (*R*)-α-methyl histamine; FST, forced swim test; SAL, saline.

**Figure 5 biomolecules-12-00683-f005:**
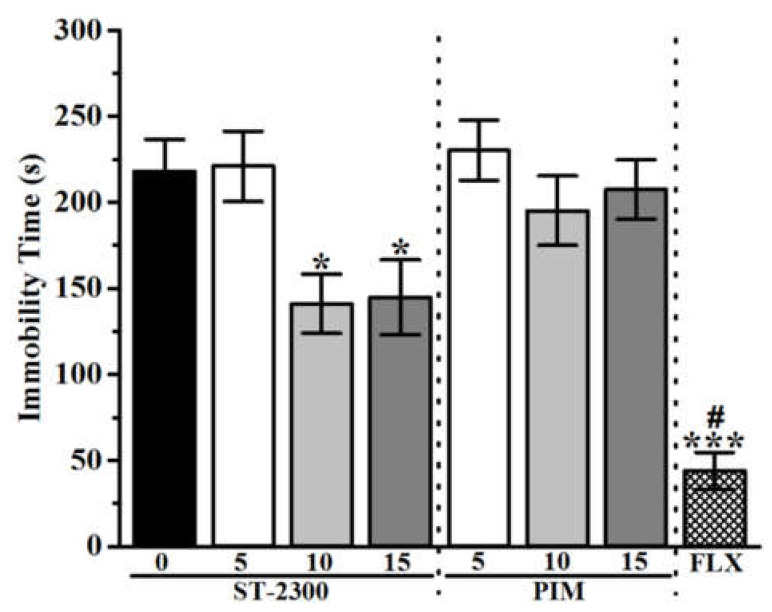
Effects of acute ST-2300 pre-treatment on depression-like behavior in TST. Acute ST-2300 (15 mg/kg, i.p.) decreased the immobility time in the tail suspension test (TST). * Denotes significant differences between ST-2300 doses and controls (*p* < 0.05); *** Denotes significant differences between FLX and saline controls (*p* < 0.001). # Denotes significant differences between FLX and administered drug doses of ST-2223 and PIM (*p* < 0.05). Vehicle (n = 9); ST-2300 (5 mg/kg, i.p.) (n = 7); ST-2300 (10 mg/kg, i.p.) (n = 6); ST-2300 (15 mg/kg, i.p.) (n = 6); PIM (5 mg/kg, i.p.) (n = 6); PIM (10 mg/kg, i.p.) (n = 7), PIM (15 mg/kg, i.p.) (n = 6); FLX (10 mg/kg, i.p.) (n = 9). Data are shown as the mean ± standard error of the mean. Abbreviations: PIM, pimavanserin; FLX, fluoxetine; TST, tail suspension test.

**Figure 6 biomolecules-12-00683-f006:**
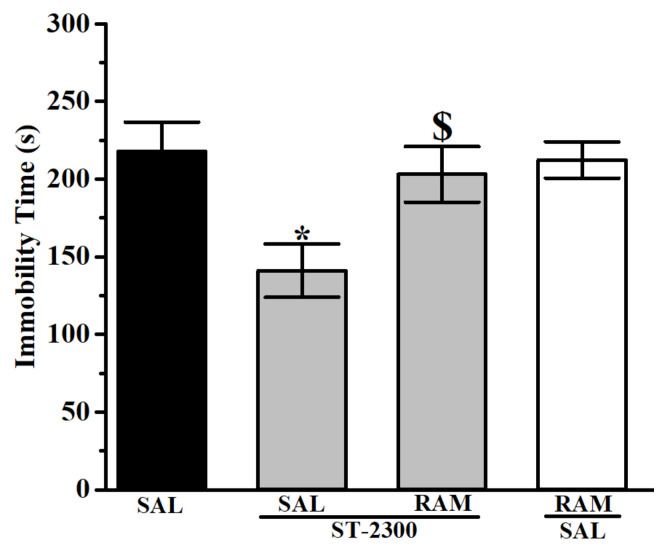
Effects of the H_3_R agonist RAM abrogated ST-2300-provided antidepressant-like effects in TST. Acute ST-2300 (10 mg/kg, i.p.) significantly decreased the immobility time in TST. * Denotes significant difference between ST-2300 (10 mg/kg, i.p.) and saline-treated control mice group (*p* < 0.05); $ Denotes significant difference when compared to ST-2300 (10 mg)-treated mice (*p* < 0.05). Vehicle (n = 9); ST-2300 (10 mg/kg, i.p.) (n = 6); ST-2300 (10 mg/kg, i.p.) co-administered with RAM (10 mg/kg, i.p.) (n = 6); RAM (10 mg/kg, i.p.) co-administered with vehicle (n = 9). Data are shown as the mean ± standard error of the mean. Abbreviations: RAM, (*R*)-α methyl histamine; TST, tail suspension test; SAL, saline.

**Figure 7 biomolecules-12-00683-f007:**
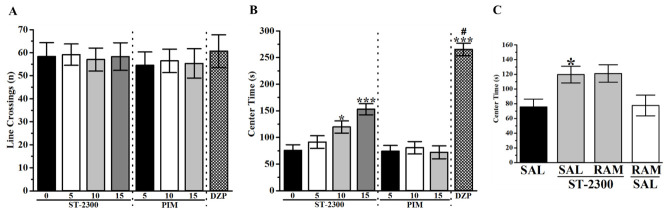
Effects of acute ST-2300 pre-treatment on anxiety-like behavior and locomotor activity in OFT. In OFT, ST-2300 had no effect on total line crossing (**A**) but dose-dependently increased the time spent in the center of the arena (**B**). Pre-treatment with the H_3_R agonist RAM did not reverse the anxiolytic effect of ST-2300 (**C**). * Denotes significant difference between ST-2300 (10 mg/kg, i.p.) and controls (*p* < 0.05); *** Denotes significant difference between ST-2300 (10 mg/kg, i.p.) or DZP (1 mg/kg, i.p.) and controls (*p* < 0.001); # Denotes significant differences between DZP and drug doses (*p* < 0.05). Vehicle (n = 9); ST-2300 (5 mg/kg, i.p.) (n = 7); ST-2300 (10 mg/kg, i.p.) (n = 6); ST-2300 (15 mg/kg, i.p.) (n = 7); PIM (5 mg/kg, i.p.) (n = 6); PIM (10 mg/kg, i.p.) (n = 7); PIM (15 mg/kg, i.p.) (n = 6); DZP (1 mg/kg, i.p.) (n = 9). ST-2300 (10 mg/kg, i.p.) co-administered with RAM (10 mg/kg, i.p.) (n = 6); RAM (10 mg/kg, i.p.) co-administered with vehicle (n = 9). Data are shown as the mean ± standard error of the mean. Abbreviations: PIM, pimavanserin; DZP, diazepam; RAM, (*R*)-α-methyl histamine; OFT, open field test; SAL, saline.

**Table 1 biomolecules-12-00683-t001:** Binding affinity or percentage inhibition of ST-2300 or PIM at different GPCRs.

Receptor Subtype	ST-2300 *K*_i_ ± SD (n) or *K*_i_ [CI 95%] (n)	Pimavanserin *K*_i_ ± SD (n)
**H_3_R**	14.4 nM [2.98 nM; 70.0 nM] (4) ^b^	inhibition at 10 µM: <50% ^a^
**5-HT_2A_**	1302 ± 331 nM (2) ^c^	0.7 ± 0.4 nM (2) ^c^
5-HT_1A_	>10,000 (2) ^d^	>10,000 (2) ^d^
5-HT_6_	>10,000 (2) ^d^	59 ± 11 nM (2) ^d^
5-HT_7_	>10,000 (2) ^d^	>10,000 (2) ^d^
**D_1_R**	>10,000 (3) ^b^	inhibition at 10 µM: <50% ^a^
**D_2_R**	>10,000 (2) ^e^	2417 ± 667 nM (2) ^e^
**D_3_R**	>10,000 (3) ^f^	inhibition at 10 µM: 60% ^a^
**H_1_R**	>5000 (2) ^f^	inhibition at 10 µM: <50% ^a^
H_4_R	>1000 (2) ^g^	n.d. ^h^

^a^ Determined by Vanover et al. [19]. ^b^ Displacement assay carried out using HEK293 cells, stably expressing the human histamine H_3_R or dopamine D_1_R against [^3^H]-*N*^α^-methylhistamine or [^3^H]-SCH23390 as radioligands, respectively. ^c^ Displacement assay carried out using CHO-K1 cells expressing the human serotonin 5-HT_2A_ receptor against [^3^H]-ketanserin as a radioligand. ^d^ Displacement assay carried out using HEK293 cell lines stably expressing the serotonin h5-HT_1A_, h5-HT_6_, or h5-HT_7_ against [^3^H]-8-OH-DPAT, [^3^H]-LSD, or [^3^H]-5-CT as radioligands, respectively. ^e^ Displacement assay carried out using HEK293 cells stably expressing the human dopamine D_2_R against [^3^H]-raclopride as a radioligand. ^f^ Displacement assay carried out using CHO-K1 cells stably expressing the human histamine H_1_R or dopamine D_3_R against [^3^H]-pyrilamin or [^3^H]-spiperone as radioligands, respectively. ^g^ [^3^H]-Histamine displacement assay performed with cell membrane preparation of Sf9 cells transiently expressing the human histamine H_4_R, co-expressed with Gα_i2_ and Gβ_1γ2_ subunits. ^h^ n.d., not determined.

## Data Availability

The datasets generated and analysed are available from the corresponding authors on reasonable request.

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
