# Peer review of "The Novel Pimavanserin Derivative ST-2300 with Histamine H3 Receptor Affinity Shows Reduced 5-HT2A Binding, but Maintains Antidepressant- and Anxiolytic-like Properties in Mice"

_biomolecules, 2022, doi:10.3390/biom12050683_

Round 1

Reviewer 1 Report

The work submitted by Venkatachalam et al. deals with the design, synthesis and in vitro and in vivo pharmacological characteristics of a new multitarget compound ST 2300,  the derivative of atypical antipsychotic drug, Pimavanserin,  with the histamine H3 receptor antagonist pharmacophore  N-(3-phenoxypropyl)piperidine substituent.

The work fits into the concept of Multi-Target-Directed Ligands,  a  modern and flowering strategy in the field of drug discovery that has replaced the former “one drug-one target” approach as most the diseases are multigenic. 

The authors hypothesized that the synergy of the effects of serotonin 5HT2a and histamine H3 receptor antagonism will be of benefit for depressive and psychiatric disorders, which are also often endowed with sleep and cognitive functions disturbance. To get confirmation, at first, the in vitro affinities of human histamine,  H1, H3, and H4 receptors,  dopamine D1, D2, and D3 receptors, and serotonin 5-HT1A, 5-HT2A, 5-HT6, and 5-HT7 receptors to ST 2300 were evaluated by relevant radioligand displacement assays using membrane preparations from appropriate cells that stably expressed respective human GPCR. As a reference, the parent compound, Pimavanserin,  was used. The ST-2300 has shown excellent binding affinity at histamine H3R of nanomolar range, however, unlike the Pimavanserin, only the moderate one at serotonin  5HT2A receptor (micromolar range) and noteworthy, none or insignificant one at the other examined monoamine receptors.

Next, preclinical animal models of depression-like behavior: forced swim test, and tail suspension test as well as anxiety-like behavior as assessed by open field test, were employed to study the in vivo effects of  ST-2300 administered to C57BL/6 mice,  and again the results were compared to those after Pimavanserin.  Acute mice pretreatment with  ST-2300 evoked antidepressant-like effects in the two tests but in the open field test, contrary to Pimavanserin,  only ST-2300 was capable of increasing the time spent in the center of the arena, without altering the overall locomotor activity of tested mice, indicating its anti-anxiety effect.

The authors concluded that the novel ligand combining  H3R and 5-HT2AR antagonist/inverse agonist affinities could be a lead compound for the synthesis of potential drugs against depression- and anxiety-related neuropsychiatric disorders.

I have some suggestions,  from the title:  

Novel Pimavanserin Derivative ST-2300 with Histamine H3 Receptor Affinity Loses 5-HT2A Binding, but Maintains Antidepressant- and anxiolytic-like Properties in Mice”  it appears, at the first glance, that there is no binding at all. However, looking at the results (Table 1) one can find the Ki is of a micromolar range,  therefore title modification is proposed. It could be f.ex.:

The Novel Pimavanserin Derivative ST-2300 with Histamine H3 Receptor Affinity Shows Weaker 5-HT2aR Binding, however, Maintains Antidepressant- and anxiolytic-like Properties in Mice

Minor: Some typo errors could be found, f.ex; line 513: orvide instead of provide,   lelels instead of levels. line 561: batary instead battery 

Author Response

  • I have some suggestions, from the title: 
    Novel Pimavanserin Derivative ST-2300 with Histamine H3 Receptor Affinity Loses 5-HT2A Binding, but Maintains Antidepressant- and anxiolytic-like Properties in Mice”  it appears, at the first glance, that there is no binding at all. However, looking at the results (Table 1) one can find the Ki is of a micromolar range, therefore title modification is proposed. It could be f.ex.:
    The Novel Pimavanserin Derivative ST-2300 with Histamine H3 Receptor Affinity Shows Weaker 5-HT2aR Binding, however, Maintains Antidepressant- and anxiolytic-like Properties in Mice

We thank the reviewer for the positive remarks as well as for the constructive comments.

Another reviewer did some negative remarks on the lenghth of the descriptive title and therefore, we would not like to extend the already ling title, but also follow the suggestion of the reviewer with a comparable title modific

Corrected: “The Novel Pimavanserin Derivative ST-2300 with Histamine H3 Receptor Affinity Shows Reduced 5-HT2A Binding, but Maintains Antidepressant- and Anxiolytic-like Properties in Mice”

  • Minor: Some typo errors could be found, f.ex; line 513: orvide instead of provide,   lelels instead of levels. line 561: batary instead battery

All typos have been corrected accordingly. Thank you for thorough reading.

Reviewer 2 Report

Depression is a heterogeneous group of disorders probably arising from various etiologies. The pathophysiology of depression takes into account genetic and environmental experiences and their interaction. Numerous biologic correlates have been identified, though none is considered causative or diagnostic. Genes that influence the production and reuptake of serotonin, norepinephrine, dopamine, and glutamate, as well as nerve cell growth in brain regions underlying memory and emotional processing, are of greatest interest. Abnormalities in brain regions underlying executive functioning, emotion regulation, and reward processing, as well as irregularities in functional connectivity have been identified. Irregularities in cortisol responding and inflammation also may play a role.

Currently, the treatment of depression is multi-directional. In human clinic the best results are achieved by the combined use of pharmacotherapy and psychotherapy.

As part of pharmacotherapy, based on the results of research on the neurochemical basis of depression, the following can be used:

  • selective serotonin reuptake inhibitor (SSRI),
  • serotonin norepinephrine reuptake inhibitor (SNRI),
  • noradrenergic and specific serotonergic antidepressants (NaSSA),
  • noradrenergic and dopaminergic antidepressants (e.g. bupropion - a selective neuronal reuptake inhibitor of catecholamines, NA and DA),
  • tricyclic antidepressants (TCA),
  • MAO inhibitors (e.g. moklobemide - belonging to the third generation of MAO inhibitors, reversibly inhibiting MAO, mainly type A; the drug inhibits the metabolism of NA, DA and 5-HT, which leads to an increase in their level in the brain; in high dose, moclobemide also inhibits MAO B).

In 2019, a novel and rapidly-acting antidepressant, esketamine (NMDA receptor antagonist), was approved by the Food and Drug Administration (FDA) for the treatment of depression in adults who have tried other antidepressant medicines but have not benefited from them (treatment-resistant depression).

The currently available pharmacological therapy does not bring any spectacular results. Moreover, the use of the above-mentioned drugs causes numerous side effects. Therefore, studies on new, safer and more effective, therapeutic strategies for depression are necessary. The manuscript of Venkatachalam et al. fits into this research trend.

Previous studies have already suggested a potential link between brain histaminergic/serotonergic signalling and antidepressant and anxiolytic effects.

The aim of the study presented in MS-1715612 was the in vivo evaluation antidepressant-and anxiolytic-like effects of the newly synthesized multiple-active ligand ST-2300. The compound  was developed from pimavanserin (an atypical antipsychotic drug, an inverse agonist of 5-HT2A and 5-HT2C receptor) and incorporates histamine H3 receptor (H3R) antagonist pharmacophore. In this way ST-2300, despite its parent compound, has only moderate serotonin 5-HT2A antagonist/inverse agonist affinities.

The H3R is distinguished for its almost exclusive expression in the CNS. It exhibits dual functionality as auto- and heteroreceptor, and this enables H3Rs to modulate the histaminergic and other cerebral neurotransmitter systems (GABAergic, glutamatergic, serotonergic, noradrenergic, cholinergic and dopaminergic).

The antidepressant-and anxiolytic-like effects of ST-2300 were evaluated using standard behavioural tests, i.e. forced swim test (FST), tail suspension test (TST) and the open field test (OFT) in adult C57BL/6 mice. Of course,  FST and TST were used to determine the antidepressant-like effects, while OFT was allowed to evaluate anxiolytic-like effects.

Fluoxetine (an antidepressant which enhances serotoninergic neurotransmission through potent and selective inhibition of neuronal reuptake of 5-HT) and diazepam (an anxiolytic drug, often used as a positive control in behavioural experiments) were used as reference compounds in the depressive-like and anxiety-like experimental model, respectively.

To check whether the observed effects were associated with antagonism of H3Rs, (R)-α-methyl histamine (RAM) was co-administrated with ST-2300. To exclude any confounding effects of the CNS-penetrant H3R agonist RAM, it was also injected to the control animals. 

The pharmacological profile of ST-2300 was also investigated using available in vitro methods.

The research methods used by the Authors of the manuscript do not raise any objections. I have the same opinion regarding the presentation of the obtained results and their interpretation.

Based on the obtained results, the Authors of MS-1715612 drew adequate conclusions. ST-2300 with H3R and 5-HT2AR antagonist/inverse agonist affinities showed promising H3R-mediated in-vivo antidepressant- as well as anxiolytic-like effects in tested animals. The Authors has well shown that multiple-active ligands with H3R affinity might address the need for safe and effective drugs with potential to be used in the treatment of depression- and anxiety-related neuropsychiatric disorders.

My questions/suggestion for Authors

As for the ability to block 5-HT2A receptors, it is also one of the pharmacological properties desired in the treatment of schizophrenia. Currently, the pharmacotherapy of schizophrenia is based primarily on the use of antipsychotic drugs combining the activity of the D2 and 5-HT2 receptor antagonists. For example, clozapine is a serotonin antagonist, with strong binding to 5-HT 2A/2C receptor subtype. It also displays strong affinity to several dopaminergic receptors with only weak antagonism at the dopamine D2 receptor. Hence, I have a question for the authors whether they take into account the assessment of the therapeutic potential of ST-2300 towards the treatment of schizophrenia.

Minor remarks - mistakes in text editing

The text of the article should be carefully traced and errors and typos should be avoided.

Examples:

line 45, 50, 57, 71, 79 - two periods instead of one

line 512 - it is necessary to verify the spacing between words; PIM (5-15 mg.kg, i.p.) - "mg" should be followed by a slash, not a period

line 441 - two parentheses before DZP

line 445 - should be "was administered"

I also propose to standardize the appearance of all figures in the article, that is, to match the appearance of figures 2 and 4 to the others. This requires, among other things, reducing the font size in the X axis description.

My critical remarks of course do not detract from the scientific value of the manuscript. After taking them into account, I recommend MS-1715612 for publication in Biomolecules.

Author Response

  • My questions/suggestion for Authors

    Hence, I have a question for the authors whether they take into account the assessment of the therapeutic potential of ST-2300 towards the treatment of schizophrenia.
  • Yes, we do consider the assessment of ST-2300 as potential drug for the treatment of schizophrenia and other related diseases. We also see that this is out of the scope of the actual paper, but we would like to address the topic in follow-up studies. We are open for collaboration.

  • Minor remarks - mistakes in text editing

    The text of the article should be carefully traced and errors and typos should be avoided.

    Examples:

line 45, 50, 57, 71, 79 - two periods instead of one.

Corrected

line 512 - it is necessary to verify the spacing between words; PIM (5-15 mg.kg, i.p.) - "mg" should be followed by a slash, not a period

Corrected

line 441 - two parentheses before DZP

Corrected

line 445 - should be "was administered"

Corrected

  • I also propose to standardize the appearance of all figures in the article, that is, to match the appearance of figures 2 and 4 to the others. This requires, among other things, reducing the font size in the X axis description.

            Corrected

Reviewer 3 Report

Dear authors,

The article "The Novel Pimavanserin Derivative ST-2300 with Histamine H3 Receptor Affinity Looses 5-HT2A Binding, but Maintains Antidepressant- and Anxiolytic-like Properties in Mice" introduces ST-2300, as a novel antidepressant and anti-anxiolytic drug. This compound was tested in vivo on mice using the forced swim test, tail suspension test, and the open field test. 

The compound was also investigated for its potential to inhibit various receptor targets: 5-HT2AR, 5-HT1AR, 5-HT6R, 5-HT7R, D1R, D2R, D3R, H1R, and H4R. The article is instructive, with well-described techniques and promising outcomes. However, I have some comments that should be included before it is published:

The title is too long; consider rewriting the title.

The abstract section is unclear, and I suggest that it be rewritten. 

Please include more data regarding the receptors studied in the introduction. The 5-HT1A receptor and the dopamine receptors are not even mentioned.

Revision of abbreviations is a general recommendation for the text. I recommend using the standard abbreviations for the receptors, e.g., 5-HT1A, not 5-HT1AR.

Why didn't you test the biological activity of the compound on 5-HTT? Most antidepressant medications primarily target this receptor. I recommend considering this.

I am not convinced that only the inhibitory activity of ST-2300 ligand on H3 and 5-HT2A receptors is responsible for the antidepressive effect. Moreover, the Ki of the ligand interacting with 5-HT2A is 1302 nM. I think this value is decent but not good enough to explain the good results obtained in in vivo tests. What Ki values were obtained between other drugs and this receptor? Please consult the literature.

Provide more information about other drugs' antidepressant effects caused by inhibition of H3 in the discussion section. 

Overall, the work is intriguing, and the findings are promising enough to serve as a starting point for future research.

However, the text is difficult to comprehend, and the paper might benefit from more clarity and explanations in the results and discussion parts.

Best regards

Author Response

  • The abstract section is unclear, and I suggest that it be rewritten.

Rewritten: “Here, we evaluated the in vivo antidepressant- and anxiolytic-like effects of the newly developed multiple-active ligand ST-2300. ST-2300 was developed from 5-HT2A/2C inverse agonist pimavanserin (PIM, ACP-103) and incorporates histamine H3 receptor (H3R) antagonist pharmacophore. Despite its parent compound, ST-2300 showed only moderate serotonin 5-HT2A antagonist/inverse agonist affinity (Ki value of 1302 nM), but excellent H3R affinity (Ki value of 14 nM). In vivo effects were examined using forced swim test (FST), tail suspension test (TST) and the open field test (OFT) in C57BL/6 mice. Unlike PIM, ST-2300 significantly increased anxiolytic-like effects in OFT without altering general motor activity. In FST and TST, ST-2300 was able to reduce immobility time similar to fluoxetine (FLX), a recognized antidepressant drug.”

  • Please include more data regarding the receptors studied in the introduction. The 5-HT1A receptor and the dopamine receptors are not even mentioned.

The focus of our in vitro characterisation was on 5-HT2A and H3 receptors. For the purpose to exclude off-target effects, the affinity to numerous other receptors was determined. In order to avoid an overly complex introduction, we refrained from describing the remaining receptors individually in the introduction. We agree that in case of a deeper focus on e.g. dopamine receptors a further description would be necessary, but this is out of the scope of this manuscript.

  • Revision of abbreviations is a general recommendation for the text. I recommend using the standard abbreviations for the receptors, e.g., 5-HT1A, not 5-HT1AR.

We have followed the suggestion of the reviewer for the 5-HT receptor subtypes, but have to state that the histamine H3 receptor (H3) can easily be misjudged to histone H3 targets. Therefore, we kept this change for the 5-HT receptor subtypes and maintained the “R” in the term of the other GPCRs tested.

Corrected

  • Why didn't you test the biological activity of the compound on 5-HTT? Most antidepressant medications primarily target this receptor. I recommend considering this.

We have focused this study on biogenic amine GPCRs. It is clear that more targets can or should be considered, e.g. transporter, ionotropic glutamate receptors etc. We hope to being able to consider testing the activity of ST-2300 on serotonin transporter and additional targets in follow-up studies which may focus on schizophrenia or related diseases.

  • I am not convinced that only the inhibitory activity of ST-2300 ligand on H3 and 5-HT2A receptors is responsible for the antidepressive effect. Moreover, the Ki of the ligand interacting with 5-HT2A is 1302 nM. I think this value is decent but not good enough to explain the good results obtained in in vivo tests. What Ki values were obtained between other drugs and this receptor? Please consult the literature.

Provide more information about other drugs' antidepressant effects caused by inhibition of H3 in the discussion section.

Added to Discussion: “Notably, previous reports in experimental rodents showed antidepressant- or anxiolytic-like effects of several few H3R antagonists/inverse agonists (e.g. ST-1283, pKi 9.62) [15–17]. However, the exact mechanisms of H3R antagonists with anxiolytic and antidepressant effects remain to be fully elucidated. Further experimental efforts are necessary to fully understand the mechanisms of histaminergic and/or other monoaminergic action involved in anxiety and depression.”

We can see that the affinities of several other ligands on these targets maybe helpful in understanding the problem. There are several excellent reviews contributing to this problems and we have cited in this manuscript only the ones, which are directly connected to the screening and discussion.